# A spatio-temporal approach to short-term prediction of visceral leishmaniasis diagnoses in India

**Emily S. Nightingale**[1]*, **Lloyd A. C. Chapman**[1], **Sridhar Srikantiah**[2],
**Swaminathan Subramanian**[3], **Purushothaman Jambulingam**[3], **Johannes Bracher**[4],
**Mary M. Cameron**[5], **Graham F. Medley**[1]

**1** Centre for Mathematical Modelling of Infectious Disease and Department of Global Health and Development, London School of Hygiene and Tropical Medicine, London, United Kingdom, **2** CARE India, Patna, Bihar, India, **3** Vector Control Research Centre, Puducherry, Chennai, India, **4** Epidemiology, Biostatistics and Prevention Institute, University of Zurich, Zurich, Switzerland, **5** Department of Disease Control, London School of Hygiene and Tropical Medicine, London, United Kingdom

* Emily.Nightingale@lshtm.ac.uk

**Data Availability Statement:** The data from the Kala-Azar Management Information System (KAMIS) underlying the results in this manuscript cannot be shared publicly because of patient

## Abstract

### Background

The elimination programme for visceral leishmaniasis (VL) in India has seen great progress, with total cases decreasing by over 80% since 2010 and many blocks now reporting zero cases from year to year. Prompt diagnosis and treatment is critical to continue progress and avoid epidemics in the increasingly susceptible population. Short-term forecasts could be used to highlight anomalies in incidence and support health service logistics. The model which best fits the data is not necessarily most useful for prediction, yet little empirical work has been done to investigate the balance between fit and predictive performance.

### Methodology/Principal findings

We developed statistical models of monthly VL case counts at block level. By evaluating a set of randomly-generated models, we found that fit and one-month-ahead prediction were strongly correlated and that rolling updates to model parameters as data accrued were not crucial for accurate prediction. The final model incorporated auto-regression over four months, spatial correlation between neighbouring blocks, and seasonality. Ninety-four percent of 10-90% prediction intervals from this model captured the observed count during a 24-month test period. Comparison of one-, three- and four-month-ahead predictions from the final model fit demonstrated that a longer time horizon yielded only a small sacrifice in predictive power for the vast majority of blocks.

### Conclusions/Significance

The model developed is informed by routinely-collected surveillance data as it accumulates, and predictions are sufficiently accurate and precise to be useful. Such forecasts could, for example, be used to guide stock requirements for rapid diagnostic tests and drugs. More

confidentiality and privacy concerns. KA-MIS data are property of the National Vector-Borne Disease Control Programme (NVBDCP, Govt of India), and are managed by CARE India. The data are available from NVBDCP (nvbdcp-mohfw@nic.in) for researchers who meet the criteria for access to confidential data. A simulated version of the dataset used in this manuscript is available at https://github.com/esnightingale/VL_prediction_paper.

**Funding:** This study was supported by the Bill and Melinda Gates Foundation (https://www.gatesfoundation.org/) through the SPEAK India consortium [OPP1183986] (ESN, LACC, SS, SS, PJ, MMC, GFM). The views, opinions, assumptions or any other information set out in this article are solely those of the authors and should not be attributed to the funders or any person connected with the funders. The funders had no role in study design, data collection and analysis, decision to publish, or preparation of the manuscript.

**Competing interests:** The authors have declared that no competing interests exist.

comprehensive data on factors thought to influence geographic variation in VL burden could be incorporated, and might better explain the heterogeneity between blocks and improve uniformity of predictive performance. Integration of the approach in the management of the VL programme would be an important step to ensuring continued successful control.

## Author summary

This paper demonstrates a statistical modelling approach for forecasting of monthly visceral leishmaniasis (VL) incidence at block level in India, which could be used to tailor control efforts according to local estimates and monitor deviations from the currently decreasing trend. By fitting a variety of models to four years of historical data and assessing predictions within a further 24-month test period, we found that the model which best fit the observed data also showed the best predictive performance, and predictive accuracy was maintained when making rolling predictions up to four months ahead of the observed data. Since there is a two-month delay between reporting and processing of the data, predictive power more than three months ahead of current data is crucial to make forecasts which can feasibly be acted upon. Some heterogeneity remains in predictive power across the study region which could potentially be improved using unit-specific data on factors believed to be associated with reported VL incidence (e.g. age distribution, socio-economic status and climate).

## Introduction

### Visceral leishmaniasis in India

The short-term forecasting of diseases targeted for elimination can be a important management tool. Visceral leishmaniasis (VL) is the acute disease caused by *Leishmania donovani*, which is transmitted through infected female *Phlebotomus argentipes* sandflies. In India, the burden of disease is largely contained within the four northeastern states of Bihar, Jharkhand, Uttar Pradesh and West Bengal, with the rural state of Bihar most broadly affected [1–3].

Incidence of VL in India has decreased substantially since the initiation of the regional Kala-Azar Elimination Programme (KEP), which aims to tackle the disease across the Indian subcontinent through enhanced case detection and treatment and reduction of vector density [4]. As a result, reported cases have fallen from 29,000 in 2010 to less than 5,000 in 2018 [3, 4]. The overall target of the programme is to reduce incidence to less than 1 case/10,000 people/year within each "block". Blocks are administrative sub-divisions of a district with population sizes varying from thirty thousand to several million, depending on geographic area and the proportion of urban and rural habitation. As a consequence, the target equates to an absolute total of between three and two hundred cases per year. To support the elimination effort, data are reported to a central repository (Kala-Azar Management Information System, KA-MIS) to construct line lists including the date and location of every diagnosed case.

Despite the overall decrease in incidence, there is considerable heterogeneity between blocks (Fig 1). In some blocks cases are now few and far between, while others remain substantially affected from year to year. The combination of the decrease and the heterogeneity raises the need for a more targeted approach; the finite resources available must be distributed efficiently to continue progress. Additionally, history has shown that VL has the potential to develop into large epidemics [5–7] and hence it is important that localised pockets of incidence

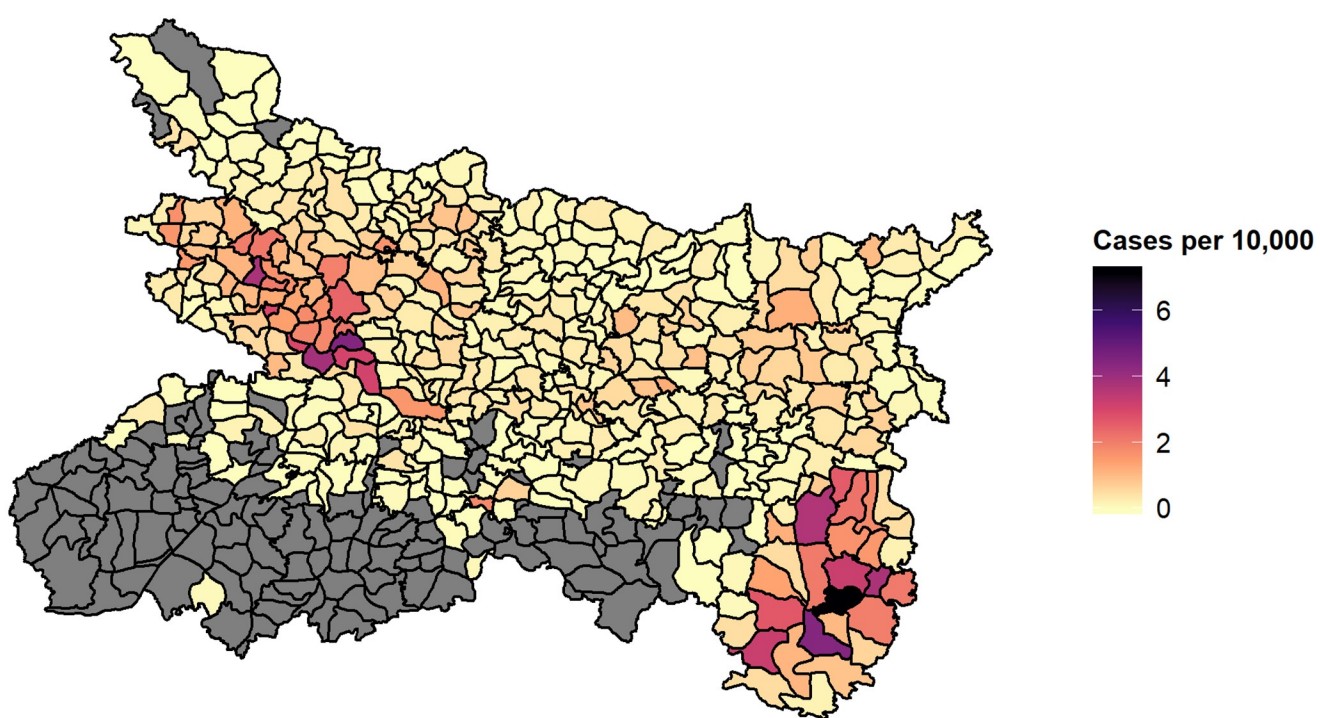

**Fig 1. Estimated incidence per 10,000 population per block in 2018, for Bihar and the four endemic districts of Jharkhand (Dumka, Godda, Sahibganj and Pakur).** Incidence is estimated according to reported cases in KA-MIS with diagnosis date in between 01/01/2018 and 31/12/2018 and block populations projected from the 2011 census according to decadal, block-level growth rates [9]. Black lines indicate block boundaries. The affected blocks of Jharkhand on average have much higher incidence than Bihar and can be seen in the bottom right of the map. Blocks marked grey had no reported cases during the study period.

are not overlooked. Intervention when incidence is low is required to prevent the trajectory from turning upwards again, as cycles of VL incidence appear to occur with a frequency of 10-20 years [8].

The primary aim of this paper is to ascertain the potential utility of predictions based solely on routinely-collected surveillance data, within a ready-made, rapid and relatively easy-to-use framework. Such predictions could serve two purposes; firstly to support logistics, for example in setting minimum stock levels of rapid diagnostic tests and drugs, and secondly to provide an early warning if the number of cases starts to resurge. For this modelling framework to be useful to the elimination programme, it is essential that its predictions are sufficiently accurate. Hence we make predictive accuracy of the forecasting approach the focus of the model selection.

## Forecasting and spatio-temporal analysis

There have been many attempts at forecasting the various forms of leishmaniasis across the three affected continents. Lewnard et al. (2014) [10] employ a seasonal ARIMA (Auto-Regressive Integrated Moving Average) model to predict cutaneous leishmaniasis in Brazil, incorporating meterological data and evaluating one, two and three month ahead forecasts. More recently, Li et al. used an extended ARIMA model to predict incidence in Kashgar prefecture, China [11]. However, neither of these attempts to capture spatial variation. Epidemiological data, in particular regarding infectious disease, are often *both* temporally and spatially correlated. That is to say, as well as incidence at one point in time being related to incidence in the

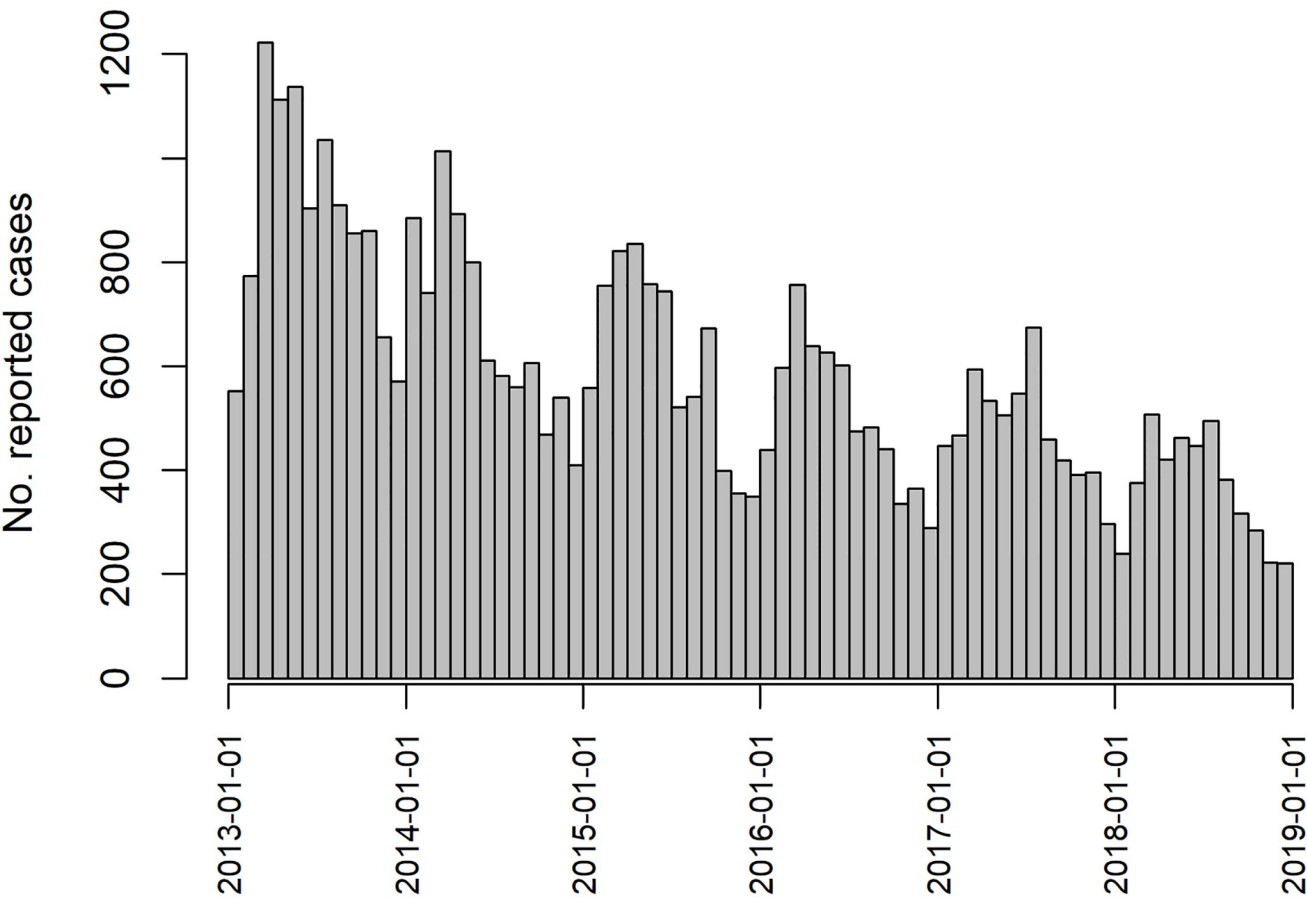

**Fig 2. Total monthly reported cases across the study region.** The annual cycle (peaking between January and April) and overall decreasing trend are clear at this aggregate level.

past, incidence in one area is also related to incidence in nearby areas. Mapping reported VL incidence in India at the block level demonstrates the presence of spatial correlation (Fig 1), with concentrated regions of high incidence appearing in West Bihar and Jharkhand. This could be due to similar geographic and demographic characteristics of neighbouring blocks, or the spread of infection by regular population movement. The seasonal cycle of incidence and overall decreasing trend (Fig 2) are clearly evident in aggregated case counts.

Several statistical approaches have been developed to model count data in space and time. These methods have been largely developed and used for understanding the drivers of patterns, often incorporating additional covariate information describing climate, geography or demography [12, 13] Dewan et al. [14] employ scanning techniques for a regional analysis solely of case data, but do not utilise the approach for prediction. Paixão-Seva et al. (2017) [15] simultaneously model the infected human, vector and dog populations in relation to landscape, climatic and economic factors, and in particular use proximity to a highway and gas pipeline as indicators of human movement. Where aetiology is not the focus, analyses often incorporate GPS locations of cases to identify hotspots and predict disease spread at a local village or household level [16], or across health facilities [17].

In the case of VL on the Indian subcontinent, environmental data are difficult to obtain in real-time at a sufficient spatial and temporal scale for forecasting purposes, and GPS data have not been routinely or uniformly collected across the affected region. As such, statistical

approaches to spatio-temporal analysis have been broadly limited to specific study regions within which additional data were collected [18]. Predictions on a regional level have so far been the remit of transmission dynamic modelling [19]. We aim to make use of the reliable and near-complete date and area data within the KA-MIS system, for the whole state of Bihar and the affected region of Jharkhand, to understand how well future cases could be predicted solely from the surveillance data of previous cases. As far as we are aware, no previous attempt has been made to forecast VL at this spatial scale and with this level of coverage for the Indian endemic region.

Often the model which best fits observed data is selected for forecasting, yet goodness of fit does not guarantee predictive power. We therefore also investigate the relationship between the fit and predictive power.

## Model framework

A natural modelling approach is to consider the cases in each month in each block as a function of cases in the previous month and in neighbouring blocks. A model framework developed in [20, 21] has been applied previously for modelling cutaneous leishmaniasis in Afghanistan [22]. This framework decomposes the distribution of counts at each point in space and time into three components (auto-regressive, neighbourhood and endemic):

- **Auto-regressive (AR)** *The contribution of previous incidence in the same block to current incidence. A choice must be made about time period of previous incidence considered (i.e. the number of months).*

- **Neighbourhood (NE)** *The contribution of previous incidence in surrounding blocks to current incidence. A choice must be made about both the time period and spatial extent considered (i.e. neighbours, neighbours of neighbours etc.), with indirect neighbours assigned decaying weights, for example, according to a power law.*

- **Endemic (END)** *A function describing the intrinsic incidence related to block factors (such as geography or demography) or seasonality.*

The sum of these components forms the mean structure for a negative binomial distribution used to model the count in each block and month. The epidemic component consists of both auto-regression and spatial/spatio-temporal regression. The maximum distance in space or time at which we assume one block-and-month count affects another is referred to as the maximum spatial or temporal *lag*. The endemic component attempts to explain any remaining variation, potentially due to overall temporal trends, population size and other unit-specific factors.

In addition to the genuine epidemiology of VL, there is an intermediary process of detection and reporting which contributes to the distribution of case counts. A new case in a previously unaffected area triggers active case detection (ACD) which continues for twelve months, therefore contributing to the pattern of temporal correlation. In other words, one case is likely to be promptly followed by more cases—not only because of transmission but also as a result of increased, localised detection effort. We therefore explored a flexible, distributed lag structure [23] which extends the range of spatio-temporal interaction by allowing incidence over multiple previous months to contribute to both the auto-regressive and spatial elements. The selection of an optimal lag length has been investigated for distributed lag models in one dimension (i.e. time alone) [24], but the impact of introducing a spatial component has not been thoroughly discussed. A strong interdependence between the autoregressive and neighbourhood components is introduced by simultaneously incorporating past information from

the same block and the neighbourhood of that block in a distributed lag model; each block affects subsequent incidence in its neighbours, which in turn affects subsequent incidence in the original block. We apply a semi-systematic approach which attempts to optimise the temporal and spatial lags simultaneously such that one does not mask the effect of the other.

## Evaluation of forecasts

The three components described in the previous section (Model Framework) have arbitrary complexity and lead to a large number of candidate models. A key issue is therefore to identify the best-fitting model, or a set of well-fitting models, and to assess to which degree good in-sample (or retrodiction) performance translates to out-of-sample forecasting performance. In-sample performance is widely assessed via the Akaike information criterion (AIC). The AIC balances the model fit and complexity, and has been recommended for model selection for prediction purposes [25]. To assess performance of probabilistic forecasts it is standard to use proper scoring rules [21, 26–29], which offer more detailed scrutiny of the prediction than measures of absolute or squared error (as used, for example, in [30]) by taking into account the whole predicted distribution. In fact, the ranked probability score (RPS) can be considered a generalisation of absolute error, to which it reduces if the forecast distribution consists of a single point. Proper scoring rules measure simultaneously the calibration and sharpness of forecast distributions; they capture the model's ability to predict both accurately and precisely but also to identify its own uncertainty in that prediction [28]. With a well-calibrated model the observed values should appear as having come from the predicted distribution at that point, and we want as precise or sharp a predicted distribution as possible while maintaining that calibration. In contrast, the mean absolute error for example only evaluates how well the central tendency of predictions aligns with the observations. We utilise the ranked probability score (RPS) [26] averaged over all predicted time points (502 blocks * 24 months, so 12048 test predictions), which for a predictive distribution $P$ and an observation $x$ is defined as

$$\overline{\mathrm{RPS}}(P, x) = \sum_{k=0}^{\infty} [F_P(k) - \mathbb{1}(x \leq k)]^2 \tag{1}$$

Here, $F_P$ is the cumulative distribution function of $P$ and 1 is the indicator function. The RPS thus compares the cumulative distribution function of $P$ to that of an "ideal" forecast with all probability mass assigned to the observed outcome $x$. We use this score rather than the logarithmic score as it is considered more robust [31], and we wish to assign some credit to forecasts near the observed value. The score is negatively oriented, meaning that smaller values are better.

Calibration can in addition be assessed using probability integral transform (PIT) histograms. The PIT histogram shows the empirical distribution of $F_{P,i}(x_i)$ for a set of independent forecasts $i = 1, \ldots, I$. We here use an adapted version for count data suggested by Czado et al [26]. If the forecasts are calibrated, the histogram should be approximately uniform. U and inverse U-shaped PIT histograms indicate that the forecasts imply too little or too much variability, respectively.

A closely-related summary measure which is easy to communicate are empirical coverage probabilities [31]. We will provide coverage probabilities of central 50% and 80% prediction intervals (reaching from the 25% to 75% and 10% to the 90% quantiles of the predictive distribution, respectively). For a calibrated forecast, the empirical coverage probabilities should be close to the nominal levels. However, in the context of sparse, low counts the discreteness of the data often prevents achieving exactly the nominal coverage level. Prediction intervals can

then either be slightly conservative (too high coverage), which is usually preferred in practice, or slightly liberal.

Our hypothesis is that models constructed with the *surveillance* framework to accommodate spatio-temporal correlation in disease incidence can provide significantly more accurate predictions (in terms of sharpness and calibration) than a purely parameter-driven (i.e. independent of history and spatial context) model with overall mean and linear time trend. Initially, we examine and discuss the relationship between model complexity, its ability to describe past data (i.e. its fit) and its ability to predict the next month. We then apply this understanding to select an optimal model for prediction with a semi-systematic approach, before comparing its predictive ability for different time horizons.

## Materials and methods

### Ethical approval

Ethical clearance was granted by the Observational/Interventions Research Ethics Committee at LSHTM (ref: 14674), subject to local approval. Local approval to use this data was granted by Dr Neeraj Dhingra, director of the National Vector Borne Disease Control Programme (GoI). Individual consent was not required as all data were analysed anonymously.

### Data

Access to the KA-MIS database of VL cases was provided by the National Vector Borne Disease Control Programme (NVBDCP) and facilitated by CARE India. Individual case records were downloaded for Bihar and Jharkhand, restricted to diagnosis date between 01/01/2013 and 31/12/2018 and then aggregated by block and diagnosis month. This gave reported case counts for 441 blocks. The KA-MIS data were merged with data from the 2011 census [9] (compiled by CARE India) for the two states to produce the final data set, including endemic blocks which had no reported cases during the study period and hence did not appear in KA-MIS. Because we incorporate spatial correlation into the model, it is necessary to not have "holes" of missing data in the map. For individual blocks within the assumed "endemic" region without any reported cases in certain months, case counts were assumed to be "true zeros" since detection efforts should be consistent with the affected neighbouring blocks. The time series for these blocks were imputed with zeros and therefore contributed to the fit of the model. Four entire districts of Bihar, at the edge of the "endemic" region, (Gaya, Jamui, Kaimur and Rohtas) had no reported cases during the period, and were excluded from the analysis.

The final analysis data set included 502 blocks across 38 districts of Bihar and Jharkhand over 72 months.

### Model structure

Due to considerable temporal variation in incidence within blocks, as a result of detection effort and cases arising in "clumps", the block-level monthly case counts are widely dispersed. A negative binomial distribution was therefore used to model the block-level case counts throughout.

All models fitted conform to the same negative binomial structure for case counts $Y_{it}$ given previous incidence:

$$Y_{it} \mid \text{past} \sim \text{NegBin}\left(\mu_{it}, \psi_i\right) \qquad (2)$$

$$\mu_{it} = \underbrace{\lambda_t \sum_{q=1}^{Q} u_q Y_{i,t-q}}_{\text{AR}} + \underbrace{\phi_t \sum_{j \neq 1} \sum_{q=1}^{Q} w_{ij} u_q Y_{j,t-q}}_{\text{NE}} + \underbrace{v_t e_{it}}_{\text{END}}. \tag{3}$$

where $Y_{it}$ denotes the reported case count in block $i$ in month $t$ with population $e_{it}$, neighbourhood weights $w_{ij}$ for neighbours $j$ of block $i$, and overdispersion parameter $\psi_i > 0$ such that $\text{Var}(Y_{it}) = \mu_{it}(1 + \psi_i \mu_{it})$. Normalised weights $u_q$ for distributed lags $q = 1, \ldots, Q$ are defined according to a scalar parameter $p$ which is estimated from the data.

$$u_q^0 = p(1-p)^{q-1}, \quad u_q = \frac{u_q^0}{\sum_{q=1}^{Q} u_q^0} \tag{4}$$

The log-transformed parameter of each model component is then defined by a linear regression on any relevant covariates, $\mathbf{X}_{it}$; in this case we consider time with sine and cosine terms to replicate seasonal waves.

$$\log(\lambda_t) = \boldsymbol{\beta}^\lambda \mathbf{X}_{it}^\lambda, \tag{5}$$

$$\log(\phi_t) = \boldsymbol{\beta}^\phi \mathbf{X}_{it}^\phi, \tag{6}$$

$$\log(v_t) = \boldsymbol{\beta}^v \mathbf{X}_{it}^v, \tag{7}$$

where $\boldsymbol{\beta}$ are the regression coefficients.

All models were fit using the R package *surveillance* [32] and its extension *hhh4addon* [33] in R version 3.6.1 (2019-07-05) [34].

**Investigating fit and prediction.** Thirty random models were drawn from the set of possible formulations (where all three of the endemic-epidemic components are included in some form) and compared on the metrics of interest. This informed the subsequent selection process for the final prediction model.

Code used to produce the results in this paper is available from https://github.com/esnightingale/VL_prediction_paper, along with a simulated version of the dataset from the final selected model.

## Model selection

During the selection process, all models were fit to the subset of months 5 to 48 in order to make comparisons between maximum temporal lags up to four months. The remaining 24 months were then predicted sequentially in a "one-step-ahead" (OSA) approach to assess predictive power (as was applied in [10]), either with rolling updates to the fit (incorporating each month's data into parameter estimates to predict the next) or without (using only the training set of data for all predictions) [22, 26]. The average RPS of these predictions served as the primary criteria for model selection, comparing between models of increasing complexity by permutation test with a significance cut-off at 0.001. At the same time, average RPS was compared to AIC from the model's training period fit to assess the relationship between fit to the "observed" data and future prediction.

The following elements were considered for inclusion in the model:

- Log of population density as a covariate in the endemic component, in place of population fraction offset.

- Seasonal variation and linear trend within the coefficients of all three components, serving to vary the relative strength of each component over time.

- Distributed temporal lags up to 4 months, with decaying weights according to a geometric distribution.

- Spatial lags up to maximum of 7th order neighbours, with weights decaying according to a power law ($w_{ij} = o_{ij}^{-d}$, where $o_{ij}$ is the neighbourhood order of blocks $i$ and $j$, and the decay exponent $d$ is to be estimated).

- Intercept of log population density in the neighbourhood component (*Gravity Law*), to reflect that blocks of high population density may be more strongly influenced by their neighbours due to migration.

- District and state-specific dispersion, allowing the variation in incidence to differ between spatial units.

It was not feasible to allow a block-specific dispersion parameter since many blocks had too few cases to obtain stable estimates.

Finer details of the model selection process are included in S1 Text.

**Empirical coverage probabilities.**   As an alternative measure of prediction utility, we calculated the empirical coverage of prediction intervals produced by each model, with respect to the observed counts. This describes the proportion of points in the test period for which the observed count fell within the middle 50% or 80% of the predicted distribution. For an ideal forecast the empirical coverage will match the nominal level. An empirical coverage probability cannot be considered "strictly proper" [21, 26, 31], as the RPS score is, and hence does not favour sharpness in addition to calibration. However, a high coverage quantile interval may provide useful lower and upper bounds for expected incidence. For more detail see S1 Text.

**Longer prediction horizons.**   For the final model, further predictions were calculated based on a rolling window of three and four months. As with the rolling OSA approach, the model was initially fit to the training set (months $1, \ldots, t$) and this fit used to predict month $t + 3$. The model was then updated with the data from $t + 1$ in order to predict $t + 4$, and so on, in a similar fashion to Lewnard et al. [10]. The RPS of one, three and four month ahead predictions were compared to assess the loss in accuracy with a longer time horizon.

## Results

*Preliminary analyses of dispersion and exploration of temporal lags are described in S2 Text.*

### Random model assessment

According to the thirty random models drawn, fit and prediction were found to be strongly correlated (Fig 3A). Predictions were calculated based on either a rolling fit (incorporating each month's data into parameter estimates to predict the next month) or fixed fit (using parameters fit to the training set only for all predictions). The scores for both prediction approaches were very similar for most models, suggesting that the processes defined in these models are consistent over time and hence the quality of prediction does not depend on regular model updates (Fig 3B). This is noteworthy since in practice it may not be possible to update the fits on such a regular basis. Selecting the model based on RPS of predictions from a fixed model fit would best reflect the constraints of reality and be the more conservative approach.

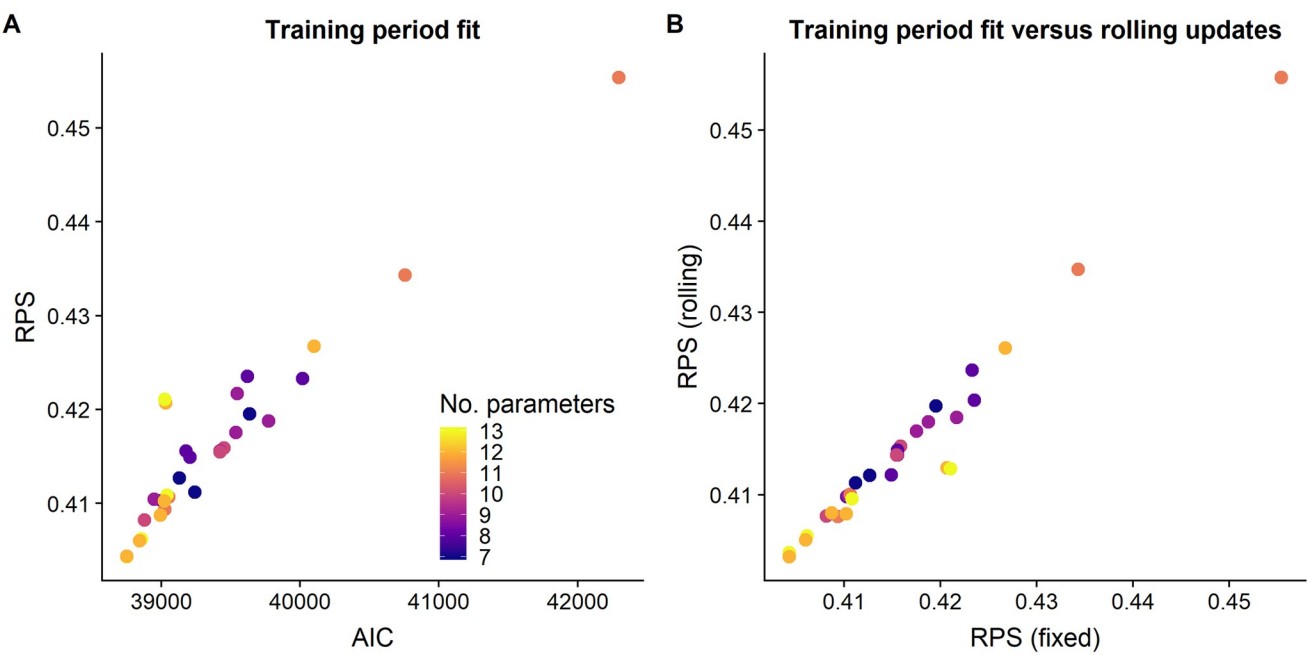

**Fig 3. Comparison of predictive performance and model fit, and predictive performance for training period fit and rolling fit updates, for models with randomly selected components.** (A) AIC versus RPS for 30 randomly selected models. AIC is calculated from the fit to the training period only (months 13 to 48) and RPS from one-step-ahead predictions (months 49 to 72) based on the same fit. According to this random sample, fit and prediction are strongly correlated; the model which fits best to the observed data produces the best one-step-ahead predictions. (B) RPS of predictions based on the fixed training set fit versus rolling fit updates. Predictive power is very similar between the two prediction approaches.

## Model selection

As was found with the random model set, the final selected model which demonstrated the highest predictive power as measured by RPS also achieved the closest fit to existing data. Initially, no more than two distributed AR lags could be added to the model without yielding evidence of miscalibration in the predictions. However, once the neighbourhood component was added in the third stage of selection, increasing the AR lags to four months significantly improved both AIC and RPS with no evidence of miscalibration. At this point the endemic linear trend lost significance and therefore was removed in subsequent models. The AIC, RPS and empirical coverage probabilities for all models considered in the selection process are shown in Fig 4. Fit and prediction metrics for all models are given in S1 Table. and PIT histograms for the models selected at each stage are compared in S3 Fig.

We found that as RPS and AIC were improved, the empirical coverage probabilities of prediction intervals were increased far beyond their nominal level. With the final model (Model no. 42), only 5.4% (652/12048) of observations fell outside the 10-90% interval, with an average interval width of just three possible case counts. This predicted distribution is much more conservative in its coverage than a simple linear trend model (coverage 10-90% = 0.905) but attains substantially better fit and RPS, suggesting that more of the improvement comes in the form of calibration. The conservative 90% predicted quantile provides a reliable upper limit for the next month's incidence, to which a management plan could be defined accordingly. The 25-75% prediction interval was found to be of limited use since, with very low counts across the majority of the region, this interval often consists of only a single value. The median would be a more interpretable value to report.

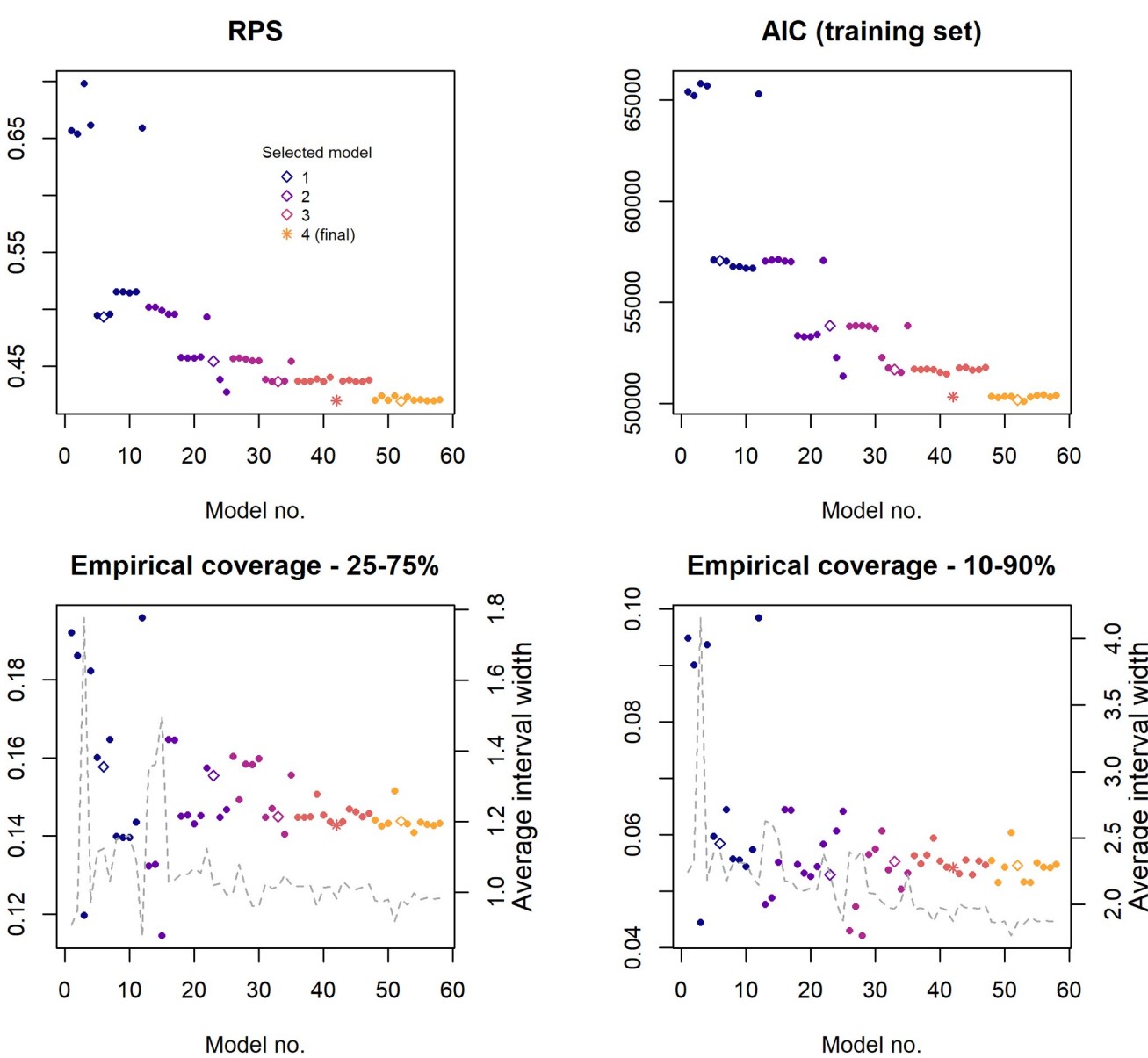

**Fig 4. Measures of fit and predictive power throughout the model selection process.** Figures illustrate the models tested in chronological order from left to right, with each stage indicated by a different colour. Models were selected at each stage based on the biggest reduction in RPS, subject to calibration; these are identified by hollow points, and the final selected model by a star. For the two variants on the coverage probability, average quantile interval width (representing uncertainty in the predicted case count) is shown on the right axis and by the grey dashed line. Interval width is determined by the count at the upper quantile minus the count at the lower, hence an interval width of two covers three possible count values (e.g. 2, 3, 4).

### Final model

The final model consists of a negative binomial distribution with a single dispersion parameter and the following mean structure:

$$\mu_{it} = \lambda_{it} \sum_{q=1}^{4} u_q Y_{i,t-q} + \phi_{it} \sum_{j \neq i} \sum_{q=1}^{4} w_{ij} u_q Y_{j,t-q} + e_{it} v_{it} \tag{8}$$

$$\log(v_{it}) = \alpha^v \tag{9}$$

$$\log\left(\lambda_{it}\right) = \alpha^{\lambda} + \gamma_1^{\lambda} \sin\left(\frac{2\pi}{12}t\right) + \delta_1^{\lambda} \cos\left(\frac{2\pi}{12}t\right) \tag{10}$$

$$\log\left(\phi_{it}\right) = \alpha^{\phi} + \gamma_1^{\phi} \sin\left(\frac{2\pi}{12}t\right) + \delta_1^{\phi} \cos\left(\frac{2\pi}{12}t\right) \tag{11}$$

The model fit is dominated by auto-regression; the majority of information with which to predict the current month comes from incidence in the previous four months, with seasonally-varying strength. Since the contribution of each component is modelled on a log scale these parameters have a multiplicative effect, hence the range of the seasonal AR component (approx. [0.6, 0.8]; see S4 Fig) indicates that each month's count is expected to be a certain fraction of the weighted average of the counts over the last four months. This occurs over all blocks and therefore amounts to an overall decreasing trend. After accounting for auto-regression, it was found that the neighbourhood effect did not extend beyond directly bordering blocks with respect to prediction. Seasonality within this component also serves to vary the magnitude of the effect throughout the year.

The contribution of an endemic trend was found to be negligible, reflecting the lack of homogeneity across blocks, and was therefore not included; the reduction in total incidence comes entirely from each block's autoregressive pattern. Block-specific covariate data (e.g. relating to socio-economic or geographic features of the area) would contribute to this component and potentially reveal associations which are consistent across blocks. Random intercepts were tested in the endemic component to capture unexplained block variation, yet did not improve predictive power in a basic model and caused convergence issues in more complex, distributed-lag models.

The relative contributions of the three model components are illustrated for the four blocks with highest average monthly incidence (Gopikandar, Kathikund, Boarijor and Sundarpahari) in Fig 5.

**Predictive performance.** The final model achieved an overall $\overline{\mathrm{RPS}}$ for one-step-ahead prediction of 0.420, 36% lower than the null (non-spatial and non-autoregressive) model and 8% lower than the best non-spatial model, with individual block-wise averages ranging from $4.3 \times 10^{-5}$ to 3.47. This equates to a mean absolute error of 0.58, a 30% reduction from the null model. That the RPS is lower than the MAE implies the probabilistic forecast is preferable to a simple point forecast.

Model selection was performed based on the model's *mean* RPS across all blocks and the whole test period but beneath this overall score is a broader distribution of scores for each block-month prediction, influenced by peaks, troughs and otherwise unusual incidence patterns. The histogram in Fig 6 illustrates the distribution over blocks, demonstrating that the final model is able to predict accurately and precisely across the majority of the region, yet there is a small subset of blocks with more widely varying RPS. It should be noted that the overall performance of the model is strongly influenced by blocks with almost no incidence as these yield the very lowest scores. Similarly, there is some correlation between the blocks for which the model performs least well, and the blocks which have historically demonstrated the highest average incidence since higher counts are harder to predict than zeros or single cases. The blocks with the highest RPS also tend to exhibit sporadic patterns or have experienced sudden, sharp changes in incidence (potentially outbreaks) within the test period, which cannot be reproduced by a model primarily informed by an average of past incidence. Examples of these patterns are illustrated in S5 Fig.

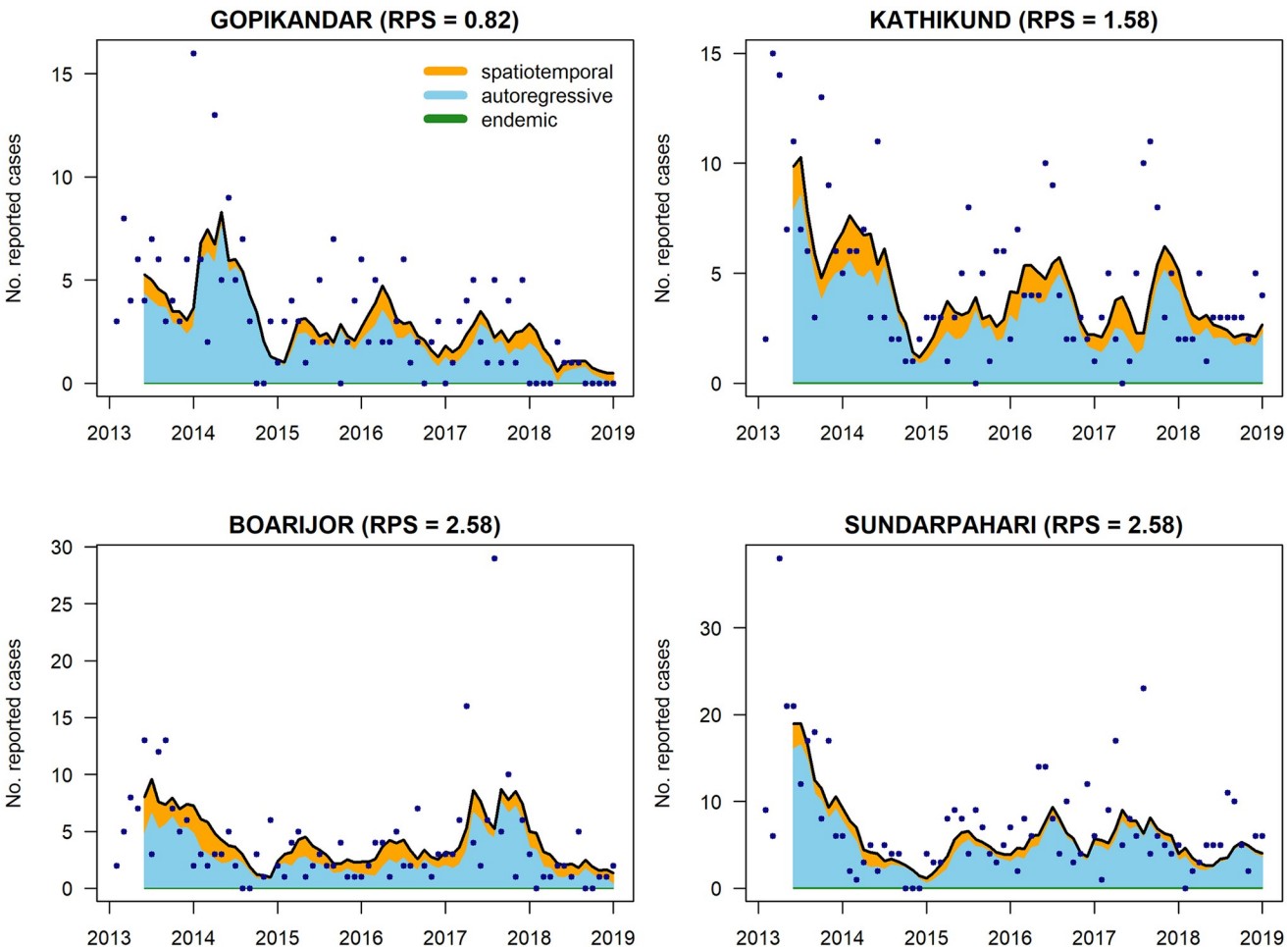

**Fig 5. Model fit for the four blocks with highest average monthly incidence (Gopikandar, Kathikund, Boarijor, and Sundarpahari, all in Jharkhand).** The observed case counts are indicated by black points and the coloured regions illustrate the relative contribution of the different model components. The contribution of the endemic component is negligible therefore barely visible. The fitted value from the model falls at the upper edge of the coloured region.

Pakur, Maheshpur, Boarijor and Sundarpahari in Jharkhand ($\overline{\text{RPS}}$ = 3.47, 2.70, 2.58 and 2.58, resp.) experienced substantial jumps in incidence between May and July 2017, constituting differences of up to 27 cases from one month to the next. Paroo ($\overline{\text{RPS}}$ = 3.07) showed a particularly erratic pattern of cases within the test period, with spikes of 21 and 19 cases separated by a few months of ∼5 cases and a subsequent fall to just one case by December 2018. Incidence in Garkha has also been inconsistent and appeared to have been on the rise in recent years, until a similar fall at the end of 2018. It should be noted that additional case detection efforts in Jharkhand at the start of 2017 will likely have contributed substantially to the observed spikes at this time.

**Three- and four-month-ahead prediction.** For the final model, further predictions were calculated based on rolling windows of three and four months. Fig 7 illustrates that the longer time window did not result in a substantial loss in predictive power, with block-wise RPS very similar for the majority of blocks. When compared over the same predicted months, the differences in $\overline{\text{RPS}}$ between one-month-ahead prediction and three-/four-month-ahead were found to be small but statistically significant (-0.024 and -0.028, resp.; p < 0.0001 for both). In terms of the empirical coverage, 85.4% of test period observations were captured in the middle 50%

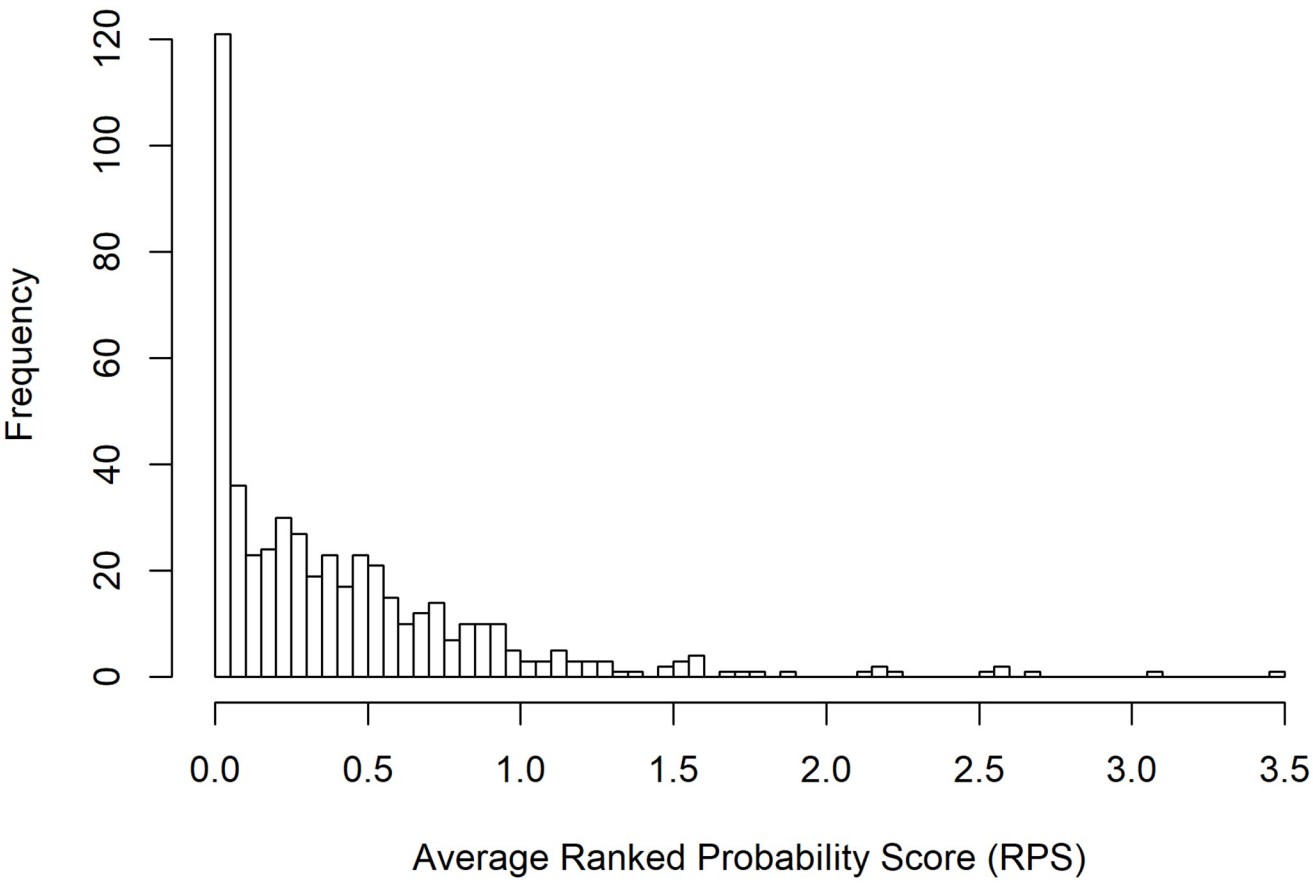

**Fig 6. Distribution of time-averaged ranked probability scores across all 502 blocks.** Low values reflect accurate and precise prediction. The majority of blocks fall below 1 with a subset for which predictive power varies widely.

of the predicted distribution based on a three month window, and 85.7% with a four month window.

Figs 8 and 9 illustrate the coverage of 45-55%, 25-75% and 10-90% prediction intervals for the block with the highest $\overline{\text{RPS}}$ of 3.47 (Pakur, Jharkhand) and a block with $\overline{\text{RPS}}$ of 1 (Bhagwanpur, Bihar). For Pakur, RPS is strongly influenced by the model's inability to match the spike in 2017, yet the incidence in surrounding months is well represented.

## Discussion

We have presented the evaluation of a predictive model of VL in Bihar and four endemic districts in Jharkhand, demonstrating a substantial (36% lower RPS) benefit from incorporating spatial and historical case information when compared to a non-spatial, linear trend model. To the best of our knowledge, this is the first time the spatio-temporal correlation of incidence at block level across all the endemic districts of Bihar and Jharkhand has been quantified. We have empirically investigated the performance of different models on prediction performance rather than model fit and produced a statistical model that is capable of accurate forecasting. Such a framework can be used as an important tool for management of endemic diseases.

Given the lack of an effective vaccine and evidence that indoor residual spraying of insecticide fails to significantly reduce sandfly densities and VL incidence in sprayed villages [35, 36],

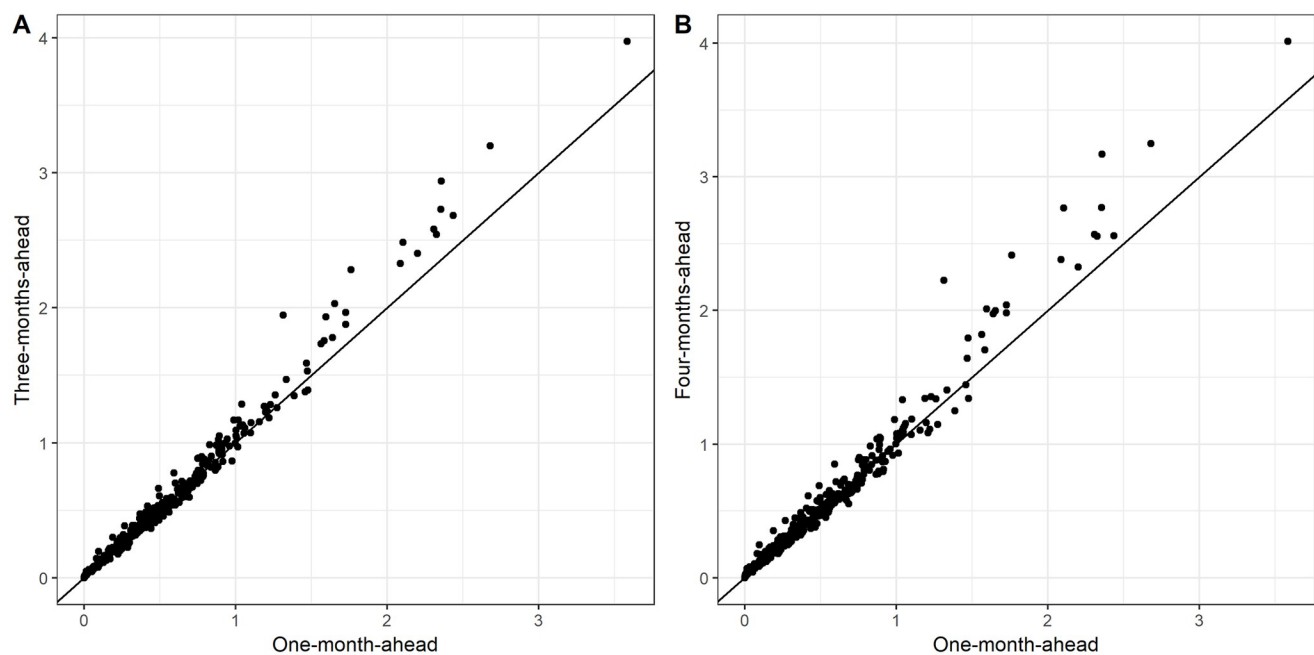

**Fig 7. Time-averaged (over months 52-72 for comparability) RPS for three- (A) and four-month-ahead (B) predictions versus one-month-ahead.**
Scores are closely matched for the majority of blocks (where $\overline{\text{RPS}} < 1.5$) but the differences increase for blocks which are harder to predict.

rapid diagnosis and treatment is currently the best method of control. With a block-level estimate of the likely number of cases to arise over the next few months, local management teams could take steps to ensure they are prepared. For example, the 90% quantile of the predicted distribution could be used to inform block-specific minimum stock levels for rapid diagnostic tests and drugs.

In practice, the prediction interval is constrained by the efficiency of the reporting process; the time taken to process diagnosis reports and input the information into the database sets a minimum horizon at which predictions would be genuinely prospective and therefore of practical use. In this paper we have assumed a delay of two months until a month's data can be considered complete, which would necessitate making predictions at least three months ahead of

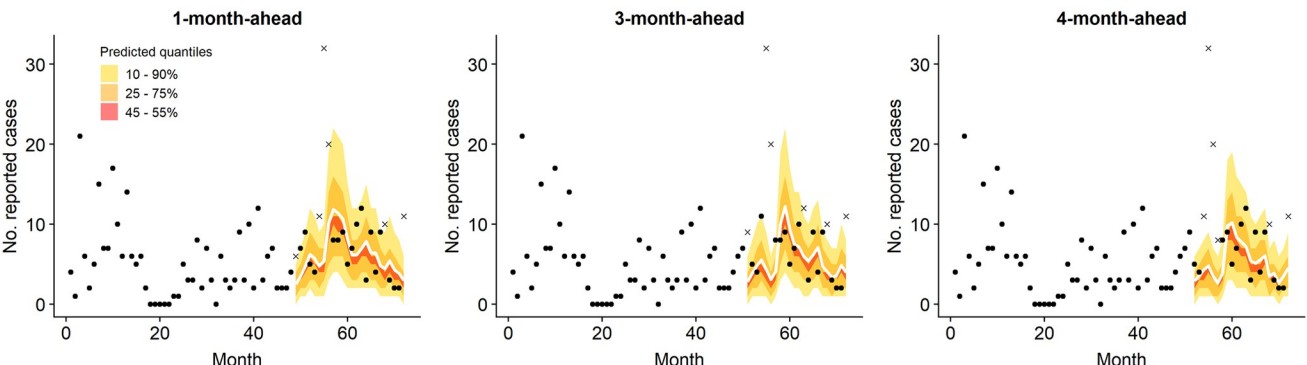

**Fig 8. One-, three- and four-step-ahead predictions (solid white line) with 10-90%, 25-75% and 45-55% quantile intervals, for Pakur block in Jharkhand ($\overline{\text{RPS}} = 3.47$ for one-step-ahead over months 49-72).** Observations which fall outside the outer prediction interval are indicated by a cross.

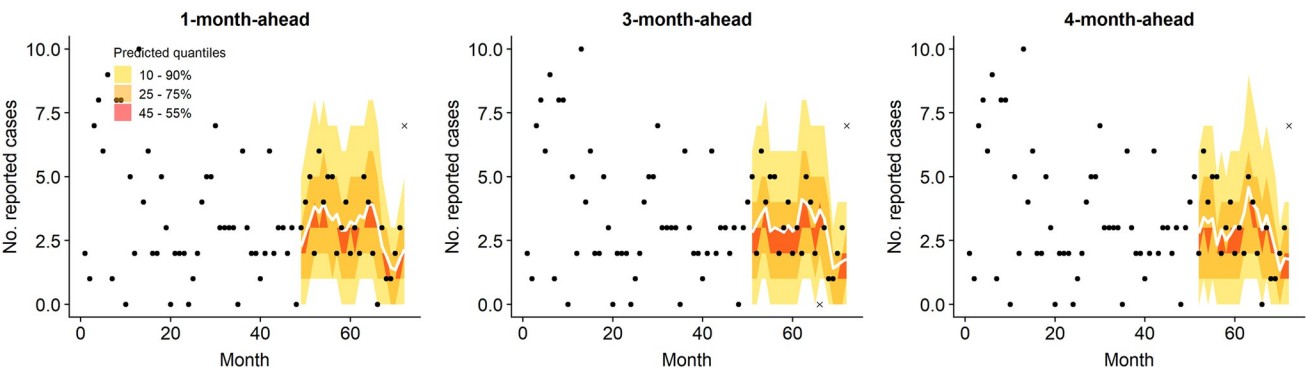

**Fig 9. Corresponding predictions for Bhagwanpur block in Bihar ($\overline{RPS}$ = 1.00).**

that point. However, conservative predictions based on preliminary month totals would still likely be of use to the national control programme.

We have demonstrated here that rolling three-month-ahead predictions are a reasonable approximation to one-month-ahead, but confidence is sacrificed for a minority of blocks as the time horizon is increased. There is a need for discussion with local disease management teams to determine the optimal balance between practicality and uncertainty with respect to predictions. Moreover, the way in which we quantify the accuracy and utility of predictions would benefit from some public health insight; it is highly likely that over- and under-estimation would need to be weighted differently, which may alter which model is deemed preferable. Ideally, the model structure would have been optimised according to predictive power on this slightly longer time horizon, but this is not a trivial task and was deemed beyond the scope of this paper.

There are also potential issues with movement of VL cases across international borders; in particular, the international boundary with Nepal cuts through a VL endemic area, artificially removing some aspects of spatial correlation. Ideally, we would take a regional perspective and also include areas in neighbouring states that have more sporadic reported VL incidence.

It could be argued that the block-level is too coarse a spatial scale for modelling the spread of an infectious disease. Outbreaks of VL occur on a smaller spatial and temporal scale than has been applied here, therefore cannot be anticipated by this model. The transmission dynamic models which are usually employed for this type of problem can be defined on a village, household or even individual level [37], yet this more detailed picture demands many more assumptions which are difficult to justify in this context. The sparseness of cases at this point in the elimination process also means that aggregation at a finer temporal scale might lead to issues with parameter estimation. The block is the unit at which control efforts are co-ordinated, disease burden is monitored, and control targets are set, therefore predictions at this level could prove to be a worthwhile compromise while more realistic transmission models are developed. With more detailed location data, the spread of disease can be modelled as a point process at the village or household level, potentially giving insight into the size and movement of disease clusters or "hot-spots" over time. This technique has previously been applied to the case of VL [38] and may be possible to extend to a larger study region in the near future, following a recent effort to collect GPS co-ordinates of affected villages across Bihar.

In this case the best-fitting model was found to be the best-predicting model. The similarity of prediction and fitting results perhaps reflects the continuity of the processes creating the

data. However, consideration of predictive power across the whole range of possible values was key to determining an optimal temporal lag length for short-term prediction. Fit and overall predictive power favoured a high number of lags in order to best capture the spatio-temporal correlation between neighbouring block counts, which appears to contribute to prediction of sudden changes in incidence. However, auto-regression is the dominant model component and appears to be captured by lags up to four months. It would be preferable to specify a different lag length for the auto-regressive and spatial components but this is not currently implemented in the *surveillance* framework. By inspection of PIT histograms, we were able to select the lag length which balanced overall predictive power with capacity to predict at the upper end of the range.

The model selection approach taken in this analysis is semi-systematic; it was not feasible to assess every possible combination of model components. Therefore we aimed to home in on a suitable model by adding components which gave the biggest improvement in predictive performance out of a range of likely options. It was found that once the major components were included in some form, further adjustment largely had the effect of redistributing the variation attributed to each component and did not substantially alter fit or prediction. There is only so much information within the time series of cases to feed the model, so predictive power quickly reaches a limit.

The analysis presented here aims to demonstrate the best that can be done with the minimal information routinely collected by the current programme, but there is evidence that this model still cannot fully account for the heterogeneity in incidence across the region. The lack of geographic and/or demographic covariates beyond population size means that the endemic component in this model is negligible; almost all our information comes from the spatio-temporal correlations, underlining the need for up-to-date data in order to make accurate predictions. Associations between VL incidence and, for example, age and socio-economic quintiles have been demonstrated [18, 39], which may give rise to varied endemic patterns at the block level. This unknown variation could in theory be quantified by random effects within this model framework, but convergence issues (likely due to the large number of zero-counts) made this infeasible in practice.

There is clearly a limitation of fitting such a model over a large number of highly heterogeneous units with minimal unit-specific information. Model selection was performed based on an average score over all blocks and time points for which predictions were made; a model is therefore chosen which predicts well overall, but in doing so sacrifices predictive power for a minority of blocks which do not follow the general trend. Zero counts dominate over all time and space, and the variance of the negative binomial distribution with a universal dispersion parameter is still too restrictive to account for blocks with the highest counts. It is in these areas where additional information on potential predictors of incidence could prove most valuable.

The variation in case counts may be better explained by a zero-inflated process, and the extent of zero-inflation will likely become more prominent as elimination is approached. Bayesian hierarchical models can be used to distinguish sources of variation at different levels and have the benefit of accommodating any informal or incomplete understanding of the transmission process within prior distributions for model parameters. These models have until recently been commonly implemented using Markov Chain Monte Carlo (MCMC) [40], which is computationally intensive for data rich in both space and time. They are however becoming increasingly accessible as a tool for inference and prediction, thanks to user-friendly wrappers which take advantage of fast computation using Integrated Nested Laplace Approximations (INLA) [41]. We will explore this approach in future work.

## Conclusion

We have demonstrated a framework for forecasting VL incidence at subdistrict level in India which achieves good predictive performance based on the available routinely collected surveillance data. This framework could be used to make short-term forecasts to provide an early indication of where case numbers are higher (or lower) than expected and to support the logistics of the elimination programme.

## Supporting information

**S1 Text. Model selection.**
(PDF)

**S2 Text. Preliminary analyses.**
(PDF)

**S1 Fig. Districts with unusual incidence patterns resulting in inflated dispersion estimates.**
(TIF)

**S2 Fig. Probability integral transform (PIT) histograms for models with increasing orders of geometric lags from 1 to 12 months (left to right, top to bottom) in the auto-regressive component.** The final model selection process considered up to four lags.
(TIF)

**S3 Fig. PIT histograms for selected models at each stage.** Model 42 is the final model. Model 52 offered minor improvement in RPS with additional complexity.
(TIF)

**S4 Fig. Fitted seasonal waves in auto-regressive (AR) and neighbourhood (NE) model components.** Both reflect the first-quarter peak in reported cases but the magnitude of the waves differs, with the contribution of the AR component varying more than that of the NE.
(TIF)

**S5 Fig. Blocks with average RPS greater than 2.5 over the test period (Jan 2017—Dec 2018).**
(TIF)

**S1 Table. Fit and prediction metrics for selected models at each stage.** The value of $S$ within the formula indicates the number of seasonal waves included. The reported AIC is for the fit to training data only, and RPS is of predictions made without updating this fit (i.e. fixed instead of rolling). C2575 and C1090 refer to the coverage of 50% and 80% quantile intervals, respectively, alongside the average interval width in cases. Model no. 42 is the final model.
(PDF)

## Acknowledgments

The authors gratefully acknowledge the support and permission of Dr Dhingra, Director of the National Vector-Borne Disease Control Programme (NVBDCP), Government of India, to use the KA-MIS data.

We thank Joy Bindroo and the staff of CARE India, Patna, for their help with data access and queries. We are also very grateful to Tim Pollington of the University of Warwick for helpful input on the *surveillance* package and model selection.

## Author Contributions

**Conceptualization:** Lloyd A. C. Chapman, Sridhar Srikantiah, Mary M. Cameron, Graham F. Medley.

**Data curation:** Emily S. Nightingale, Sridhar Srikantiah.

**Formal analysis:** Emily S. Nightingale, Lloyd A. C. Chapman, Graham F. Medley.

**Funding acquisition:** Mary M. Cameron, Graham F. Medley.

**Investigation:** Sridhar Srikantiah.

**Methodology:** Emily S. Nightingale, Lloyd A. C. Chapman, Johannes Bracher, Graham F. Medley.

**Project administration:** Graham F. Medley.

**Resources:** Mary M. Cameron, Graham F. Medley.

**Software:** Emily S. Nightingale, Johannes Bracher.

**Supervision:** Lloyd A. C. Chapman, Graham F. Medley.

**Validation:** Emily S. Nightingale.

**Visualization:** Emily S. Nightingale.

**Writing – original draft:** Emily S. Nightingale.

**Writing – review & editing:** Emily S. Nightingale, Lloyd A. C. Chapman, Swaminathan Subramanian, Purushothaman Jambulingam, Johannes Bracher, Graham F. Medley.

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
