## [Decision Letter · Decision Letter 0]

18 Feb 2020

Dear Ms Nightingale,

Thank you very much for submitting your manuscript "A spatio-temporal approach to short-term forecasting of visceral leishmaniasis diagnoses in India" for consideration at PLOS Neglected Tropical Diseases. As with all papers reviewed by the journal, your manuscript was reviewed by members of the editorial board and by several independent reviewers. In light of the reviews (below this email), we would like to invite the resubmission of a significantly-revised version that takes into account the reviewers' comments. I acknowledge that the reviewers' comments may seem somewhat contradictory -- however, strive to answer the reviewers' comments inasmuch as possible.

We cannot make any decision about publication until we have seen the revised manuscript and your response to the reviewers' comments. Your revised manuscript is also likely to be sent to reviewers for further evaluation.

[1] A letter containing a detailed list of your responses to the review comments and a description of the changes you have made in the manuscript [please specify page and line numbers where changes have been made in the revised manuscript]. Please note while forming your response, if your article is accepted, you may have the opportunity to make the peer review history publicly available. The record will include editor decision letters (with reviews) and your responses to reviewer comments. If eligible, we will contact you to opt in or out.

Sincerely,

Richard Reithinger

Associate Editor

Charles Jaffe

Deputy Editor

Reviewer's Responses to Questions

**Key Review Criteria Required for Acceptance?**

**Methods**

-Are the objectives of the study clearly articulated with a clear testable hypothesis stated?

-Is the study design appropriate to address the stated objectives?

-Is the population clearly described and appropriate for the hypothesis being tested?

-Is the sample size sufficient to ensure adequate power to address the hypothesis being tested?

-Were correct statistical analysis used to support conclusions?

-Are there concerns about ethical or regulatory requirements being met?

Reviewer #1: The objectives of the study are not clearly articulated as the manuscript read like a tutorial for using spatio temporal models to produce and evaluate forecasts.

The study fails to properly address the previous literature on leishmaniasis forecasts, for example (just looking at PLoS NTDs):

https://journals.plos.org/plosntds/article?id=10.1371/journal.pntd.0003283

https://journals.plos.org/plosntds/article?id=10.1371/journal.pntd.0005353

https://journals.plos.org/plosntds/article?id=10.1371/journal.pntd.0000033

No clear hypothesis is being tested about the forecasts, for example, a relevant question not addressed is how would nested simplified models perform when compared with the best model in the s1 table. To what extent is the spatial or temporal component really necessary for an accurate prediction? This is related with the issue that model search was not fully systematic (see lines 464-471). Authors could have tried model selection, at least, about the local AIC minima and the models with best prediction performance. 

Methodologically, it is not clear why authors included time in both the temporal and spatial parts of the model. Related to this, it is also unclear why the authors try to propose new terms (autocorrelation, neighborhood and endemic) for what has been traditionally referred as temporal, spatial and external (also exogenous or extrinsic) in the time series literature and in the theory for nonlinear dynamic models (e.g., Kaplan and Glass 1997) that underlies the type of model fitted.

Related the temporal auto-correlation the selection of sine and cosine functions is known to be prone to grossly misrepresent the seasonality of environmentally forced diseases, as seasonal patterns tend to be asymmetric and are more clearly described by seasonal autoregressive functions or via the decomposition of signals (See Priestley 1988, https://books.google.co.cr/books/about/Non_linear_and_Non_stationary_Time_Serie.html?id=DvywVGExSZkC&redir_esc=y). 

In the models it is not clear why time was included both in the temporal and spatial part of the model, using the same functions and what was done to avoid problems of unidentifiability.

The part about model evaluation is also written in a highly mystified way, since at the end what is done testing whether the model predictions capture the peaks and throughs in cases.

In line 115 it is worth checking that scoring rules were used in the suggested references above.

In Lines 196-200 the comparison of rolling windows has been previously used in leishmaniasis forecasts (see suggested references above)

Reviewer #2: With this paper being about the development and assessment of a model, the methods (with ample information needed from the background section) well described the model development process and implementation for developing potential models to compare for the best potential predictor. This was very mathematically complicated, but I think covered well their development of the model. There was no stated "hypothesis" per se, but essentially they are looking at among the models made, what is the best model to predict future cases. Study design is appropriate to this model development, the population and data set were clearly described with the "sample size" of blocks being clearly described and justified (no clear sample size calculation, seems to be based on availability rather than statistically driven number of blocks needed to make the best model; while perhaps not ideal, this is a real-world application and I think appropriate methodology was used in this paper). No concern about the statistical tests to compare the models as described. No concerns regarding the ethical requirements.

**Results**

-Does the analysis presented match the analysis plan?

-Are the results clearly and completely presented?

-Are the figures (Tables, Images) of sufficient quality for clarity?

Reviewer #1: The lack of reference to previous studies makes difficult the comparison of the forecasts, since measurements like the reduction in the mean square error (or predictive r squared) are not presented.

lines 252-256: Seems it will be best to try zero inflated count models for those districts.

Table s1 only shows what seems like a rather small subsample of the models explored (looking at the model numbers).

Reviewer #2: The analysis matches what was planned in methods well. The results are described well in the text with graphics to help illustrate (though some of these are less helpful to the average reader and could be made supplemental (such as Figures 5 and 6). Overall very readable for a very densely mathematical paper. The authors start to overlap into discussion a little in the results section (e.g., talking about limitations of the models produced).

**Conclusions**

-Are the conclusions supported by the data presented?

-Are the limitations of analysis clearly described?

-Do the authors discuss how these data can be helpful to advance our understanding of the topic under study?

-Is public health relevance addressed?

Reviewer #1: lines 282-285, this conclusion might be an artifact of the non-systematic model selection.

Reviewer #2: Conclusions are supported by the data, and between here and the results section the limitations of the model are well described (such as model driven by low incidence blocks and that areas with higher cases or sporadically high case burden were less well predicted). They also very clearly stated that this model would be limited if incidence continues to fall off and that the approach needs to be broadened to a more regional approach that is not artificially bounded by country borders. They very practically designed the model to meet the needs of the elimination program with consideration of data delay built in trying to balance practicality with predictability. This is very relevant and applicable to the real world situation program in Southern Asia.

**Editorial and Data Presentation Modifications?**

Reviewer #1: Be more careful when compiling your latex files in line 209 (and others) the "<" was written as ">"

Reviewer #2: Overall very well written and understandable language used.

**Summary and General Comments**

Reviewer #1: This study is on an interesting topic, though it shows very preliminary results and fails to put in context the results in light of what has been done previously for leishmaniasis forecasting. Surprisingly no effort was made to test whether meteorological information improved the forecasts. One major issue is that the analysis was very constrained based on what was implemented in the R package used to fit the models. 

The study uses a relatively novel way to study time series, yet the methods are described in a highly mystified way, for the general audience of PLoS NTDs.

Reviewer #2: Overall very good paper with an aim to be practical. As above there are a few more technical figures that can be added to the supplementary material and some of the results would be more ideally incorporated in the discussion as many readers practically will read the abstract then the discussion and then if interested go back to the results and maybe the methodology so for those readers they will miss some of the very valuable discussion on the limitations and why those were chosen and the practical approach the authors took to develop this model. Just a suggestion though as overall a very good paper.

PLOS authors have the option to publish the peer review history of their article (what does this mean?). If published, this will include your full peer review and any attached files.

Reviewer #1: No

Reviewer #2: Yes: Nathanial K. Copeland
---

## [Editor Report · Decision Letter 1]

24 May 2020

Dear Ms Nightingale,

We are pleased to inform you that your manuscript 'A spatio-temporal approach to short-term forecasting of visceral leishmaniasis diagnoses in India' has been provisionally accepted for publication in PLOS Neglected Tropical Diseases.

Best regards,

Richard Reithinger

Associate Editor

Charles Jaffe

Deputy Editor

---

## [Editor Report · Acceptance letter]

25 Jun 2020

Dear Ms Nightingale,

We are delighted to inform you that your manuscript, "A spatio-temporal approach to short-term forecasting of visceral leishmaniasis diagnoses in India," has been formally accepted for publication in PLOS Neglected Tropical Diseases.

Best regards,

Shaden Kamhawi

co-Editor-in-Chief

Paul Brindley

co-Editor-in-Chief
